# Assessment of Spatio-Temporal Land Use/Cover Change and Its Effect on Land Surface Temperature in Lahaul and Spiti, India

**Md. Arif Husain [1], Pankaj Kumar [2,*] and Barbaros Gonencgil [3]**

1 Department of Geography, Shyama Prasad Mukherji College for Women, University of Delhi, New Delhi 110026, India
2 Department of Geography, Delhi School of Economics, University of Delhi, New Delhi 110007, India
3 Department of Geography, University of Istanbul, Istanbul 34452, Turkey
* Correspondence: pankajdsedu@gmail.com or pkumar@geography.du.ac.in

**Abstract:** Land Use/Land Cover (LULC) changes have a significant impact on Land Surface Temperature (LST). The LST is an important parameter in various environmental and climatological studies, as it plays a crucial role in understanding the Earth's surface–atmosphere interactions. The LULC changes can modify the surface energy balance and alter the radiation budget, leading to changes in LST. Urbanization, deforestation, and agricultural land use changes are some of the primary drivers of LULC change that have a significant impact on LST. Deforestation and agricultural land use changes result in a reduction in evapotranspiration, leading to an increase in LST. The main objective of the study is to analyze the spatio-temporal change in Land Use/Land Cover (LULC) and its effect on Land Surface Temperature (LST), as well as to establish a correlation of LST with the Normalized Difference Vegetation Index (NDVI) and Normalize Difference Snow Index (NDSI). Understanding the impact of LULC on LST is essential for developing effective land use policies that can mitigate the adverse effects of LULC change on the environment and human health.

**Keywords:** Lahaul and Spiti; Landsat; LULC; LST; NDVI; NDSI





## 1. Introduction

The change in land use and cover is the major form of environmental change that occurs in the Himalayan region [1]. The transformation of land is caused by natural driving forces and anthropogenic activities. The key issue is its impact on the regional environment in understanding the relationship between society and the environment [2–4]. Land use and land cover are two different terminologies, where land cover refers to everything designed by nature on the earth's surface and land use indicates the utilization of land cover for different purposes [5]. Change in LULC is dynamic, driven by natural processes and manmade activities which impact the natural ecosystem [6]. LULC change is the single most important indicator of global change which is associated with climate change. LULC change has a significant impact on the environment [7]. Land use/cover change does not always indicate land degradation. Several studies conducted on the LULC change in the western Himalayas indicated that major change occurred in agriculture, grassland, settlement, barren land, snow cover, and salix plantation, etc., over the last 2–3 decades [8,9]. Land use is the management and modification of land to utilize and capture the land cover through anthropogenic activities such as agricultural fields, residential areas, grazing, mining, and logging, etc. [10]. LULC change in the western Himalayas is an important issue due to the region's ecological, economic, and cultural significance. The Himalayan region is experiencing rapid changes in land use and land cover due to various drivers such as population growth, climate change, and developmental activities [11–13].The land use/land cover pattern of any region is the outcome of socio-economic factors and nature, and their use by human beings over space and time. The detailed information of land use/cover (LULC) and its optimum use is important for the selection, planning, and

implementation of land use schemes to fulfill the increasing demand of society [14]. Land use/land cover change has a great influence on the earth's system including land surface temperature (LST), climate, and hydrology [15].

Land Surface Temperature (LST) is the radiative skin temperature of the earth which is received from solar radiation [16]. It depends on the LULC types and the amount of sunlight received by the earth's surface. It is one of the most important indicators of the energy balance on the earth and key parameters of microclimate study [17,18]. The main objective of the present study is to analyze the effects of Land Use/Land Cover Change (LULC) on Land Surface Temperature (LST) in the Lahaul and Spiti district. The study is primarily based on Landsat satellite dataset. Remote sensing data have been widely used to assess and examine changes in the environment such as change in LULC, LST, forest area, fresh water, and agricultural patterns [19–21].

The Lahaul and Spiti is a cold desert mountain, situated in the northern part of Himachal Pradesh, India [22]. The terminology desert is used to describe the study area, indicating that it is already scarce in natural resources, which are essential for a life support system. The region is sensitive to anthropological activities and climate change. The studies suggest that LULC change in Lahaul and Spiti is mainly driven by agricultural expansion, urbanization, and developmental activities, and has significant implications for the region's ecology, livelihoods, and traditional way of life [23–26]. A rapid pace of developmental activities and making the area easily accessible to mass population change the natural LULC setup. The change in LULC induced the land surface temperature change, which has multiple adverse effects in Lahaul and Spiti.

## 2. Study Area

Lahaul and Spiti district is a sparsely populated situated in Indian state of Himachal Pradesh. The district has tough terrain and consists of two great valleys, Lahaul and Spiti. The headquarters of the district is at Keylong, which lies between 31°44′57″ and 32°59′57″ N latitudes and 76°46′29″ and 78°41′34″ E longitudes. It is covered by the Survey of India sheets 52C, 52D, and 52L. It is surrounded by Jammu and Kashmir in the north, Tibet in the east, Kulu in the south, Kinnaur in the southeast, and Kangra in the northeast (Figure 1).

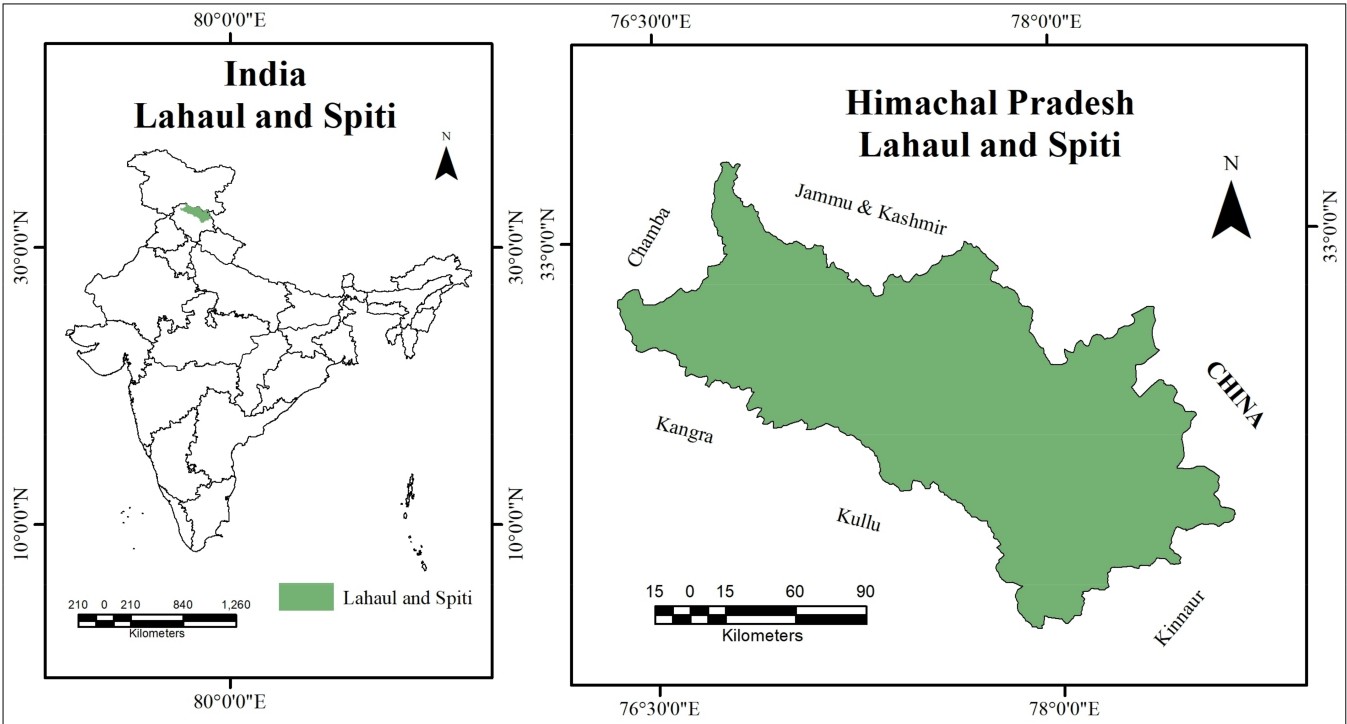

**Figure 1.** Study Area.

The district has 521 villages, where only 287 villages are inhabited and 234 are uninhabited. It has been divided into two divisions Keylong and Kaza. The district has two tehsils (Kaza and Keylong) and one sub-tehsil (Udaipur). It has a population of 31,564 with density of 2 persons per km$^2$ as per the census 2011, ranked 12th in the state on the basis of population. The sex ratio of the district is 903, in which the male and female populations are 16,588 and 14,976, respectively.

## 3. Data Base

Landsat imagery is the primary data source for examining land use/land cover (LULC) and land surface temperature (LST) changes. Landsat data of the post-monsoon season are used to avoid cloud cover and for better analysis. Data of the month of September are downloaded from the NASA through its Earth Explorer (USGS) data portal. Landsat imagery includes Landsat the Operational Land Imager/Thermal Infrared Sensor (OLI/TIRS) and Thematic Mapper (TM) scenes of 1990, 2015, and 2020. In addition, the Aster Digital Elevation Model (DEM) is used to analyze changes in LULC and LST as a function of altitude zones. The summary of the Landsat images used in the study is given in (Table 1). The satellite images of the same month in 1990, 2015, and 2020 were used to reduce the effect of seasonal discrepancies on the classified result. The data were reprojected onto the 43N zone of the Universal Transverse Mercator (UTM) projection system and the World Geodetic System 84 (WGS 84) datum, ensuring accuracy between datasets during analysis. Pre-processing of satellite data (radiometric, atmospheric, geometric corrections, and haze reduction) was performed using ArcMap 10.5 and Erdas Imagine 2015 software.

**Table 1.** Data sources: Earth-explorer, USGS.

| Satellites | Raw | Path | Date of Acquisition | Scene Centre Time | Bands | Sensor | Resolution |
|---|---|---|---|---|---|---|---|
| Landsat 5 | 37, 38 | 147, 148 | 10 September 1990 | 04:38:03 | 4, 3, 2 | TM | 30 m |
| Landsat 5 | 37, 38 | 147, 148 | 10 September 1990 | 04:20:45 | 6 | Thermal | 120 m × 30 |
| Landsat 5 | 37, 38 | 147, 148 | 10 September 2010 | 04:38:03 | 4, 3, 2 | TM | 30 m |
| Landsat 8 | 37, 38 | 147, 148 | 5 September 2020 | 05:17:55 | 5, 4, 3 | OLI | 30 m |
| Landsat 8 | 37, 38 | 147, 148 | 5 September 2020 | 05:30:20 | 10, 11 | TIRS | 100 m × 30 |

Source: Landsat, USGS (Earth Explorer).

The temperature data of the month of September were acquired from the NASA Langley Research Center POWER Project funded through the NASA Earth Science Directorate Applied Science Program. The air temperature was recorded at a height of 2 m from the ground. The specific six points were entered in the POWER interface within the district boundary of Lahaul and Spiti with the decimal degrees coordinates of Tabo (latitude 32.1026 and longitude 78.4867), Dhankar (latitude 32.1328 and longitude 78.4416), Kungri (latitude 31.9428 and longitude 77.9948), Kaza (latitude 32.5733 and longitude 77.0322), Chhota Dara (latitude 32.2772 and longitude 77.4269), Keylong (latitude 32.228 and longitude 78.0517), Udaipur (latitude 32.7283 and longitude 76.6709), and Triloknath (latitude 32.7372 and longitude 76.4488).

## 4. Methodology

### 4.1. Computation of Land Use/Land Cover Change

There is no perfect classification of land use and land cover, and it is unlikely that it will ever be developed. There are different views on the classification process, and even when using a numerical approach, the process itself is often subjective. In fact, as patterns of land use and land cover change due to natural demands and degradation, there is no logical reason to expect a detailed inventory to be sufficient for more than a short period of time. In the present study, the supervised maximum likelihood classification technique is used to examine the changes in LULC. It is an important tool that is used to extract

information in numerical values from remote sensing satellite image data. The supervised classification is performed using the composition of bands of blue, green, and red using the False Color Composite (FCC) to select the region of interest for features such as water body, vegetation, dry land, and snow, etc., in Erdas Imagine 2015. Approximately 2500 to 3000 pixels are taken from each category as samples of the five attributes during image classification. These selected pixels are called spectral signature in the field of remote sensing technology. The spectral signature of each class is obtained from the reference data and is performed by pixel selection for each of the LULC types. These pixels help develop the image by identifying an area of the image based on the color assigned to that category and the spectral homogeneity of the pixels in a particular area. The different types of LULC are grouped into five classes for efficient analysis and easy change detection assessment. This technique is used because more than five LULC classifications would not be possible with the 30 m resolution of satellite images. The detailed summary of LULC categories has been given below in Table 2.

**Table 2.** Land Use/Cover Types.

| LULC | General Description |
|---|---|
| Sparse Vegetation | Small shrubs and natural grass |
| Dense Vegetation | Crops, apple orchards and trees |
| Snow | Area covered with snow |
| Barren Land | Bare exposed rocks |
| Water Body | River, stream, glacial lake, etc. |

Source: Compiled by Author.

LULC categories include snow cover, barren land, dense vegetation, sparse vegetation, and water bodies. The water bodies include rivers, streams, glacial lakes, water-logged areas, and small ponds. The barren land category includes fallow land, waste land, bridges, settlements, and roads, etc. Dense and sparse vegetation includes trees, agricultural fields, apple orchards, grass, and shrubs. Snow cover and barren land are the dominant categories among all other LULC types.

*4.2. Computation of NDSI and NDVI*

- Normalized Difference Snow Index (NDSI) is computed by using Visible Infrared Sensor (VIS), Short-Wave Infrared (SWIR), and bands of Landsat data [27,28].

$$NDSI = (Green - SWIR)/(Green + SWIR) \tag{1}$$

- Normalized Difference Vegetation Index (NDVI) is computed by using Near-Infrared and RED Sensor [29].

$$NDVI = (NIR - RED)/(NIR + RED) \tag{2}$$

*4.3. Estimation of Land Surface Temperature (LST)*

Land surface temperature is determined by using the established equation based on Liqin et al. 2008 [30]. The methodology has been explained below to estimate brightness temperature from the thermal band of Landsat Images.

- Top of Atmosphere (TOA) Radiance: Thermal infrared digital number (DN) is converted into spectral radiance (L)

$$L(\lambda) = (L_{MAX} - L_{MIN})/255 \times DN + L_{MIN} \tag{3}$$

where
$L(\lambda)$ = Spectral radiance;

$L_{MIN}$ = Spectral radiance of DN value;
$L_{MAX}$ = Spectral radiance of DN value;
DN = Digital Number.

- Top of Atmosphere (TOA) Brightness Temperature: Spectral radiance is converted into top of atmosphere brightness temperature in Kelvin.

$$T = K2/In\ (K1/R) + 1 \tag{4}$$

where
K1 = Calibration Constant 1 (607.76);
K2 = Calibration Constant 2 (1260.56);
R = Radiance values $W/m^2$ SR μm;
T = Surface Temperature (in Kelvin).

- Land Surface Emissivity (LSE) is derived with the help of NDVI value using the formula which is given by Sobrino and Raissouni [31].

$$LSE = 1.0094 + 0.047 \times Ln\ (NDVI) \tag{5}$$

- Finally, Kelvin is converted into degrees Celsius with the equation given below:

$$TB = T - 273 \tag{6}$$

### 4.4. Mann–Kendall Test

The Mann–Kendall test is a non-parametric test for identifying trends in time series data. The test compares the relative magnitudes of sample data rather than the data values themselves [32]. The Mann–Kendall test uses two terminologies, H0 and H1, where H0 assumes that there is no trend and hypothesis is null (independent and randomly distributed data). The alternate hypothesis H1 assumes that there is a trend. Hence, H0 is tested against H1. The Mann–Kendall test statistic S [33,34] is calculated as follows:

$$S = \sum_{k=1}^{n-1} \sum_{j=k+1} \text{sign}(x_j - x_k)$$

$$\text{sign}(x_j - x_k) = \begin{cases} 1 \text{ if } x_j - x_k > 0 \\ 0 \text{ if } x_j - x_k = 0 \\ -1 \text{ if } x_j - x_k < 0 \end{cases} \tag{7}$$

A very high positive value of *S* is an indicator of an increasing trend, and a very low negative value indicates a decreasing trend. The Mann–Kendall test was computed with help of software XLSTAT 2021. The statistical significance of a trend is evaluated using the Z value.

The variance is computed using the following equation:

$$Var(S) = \frac{n(n-1)(2n+5) - \sum_{i=1}^{m} t_i(t_i - 1)(2t_i + 5)}{18} \tag{8}$$

where $n$ is the number of data points, $m$ is the number of tied groups, and $t_i$ denotes the number of ties of extent $i$. A tied group is a set of sample data having the same value.

### 4.5. Estimation of Sen's Slope

Sen (1968) developed the non-parametric procedure for estimating the slope of trend in the sample of N pairs of data [35]:

$$Q = \text{Median}\ (x_j - x_k/j - k)\ j > k \tag{9}$$

where $x_j$ and $x_k$ are the data values at times j and k (j > k), respectively. If there is only one datum in each time period, then N = n(n − 1)/2, where n is the number of time periods. If there are multiple observations in one or more time periods, then N < n(n − 1)/2, where n is the total number of observations. The N values of Qi are ranked from smallest to largest and the median of slope or Sen's slope estimator is computed as

$$Q_{med} = \begin{cases} Q_{[(N+1)/2]}, & \text{if } N \text{ is odd} \\ \dfrac{Q_{[N/2]} + Q_{[(N+2)/2]}}{2}, & \text{if } N \text{ is even} \end{cases} \tag{10}$$

The $Q_{med}$ sign indicates data trend reflection, while its value indicates the steepness of the trend. To determine whether the median slope is statistically different than zero, one should obtain the confidence interval of $Q_{med}$ at a specific probability.

In the present study, Sen's slope non-parametric method was used to estimate the slope of existing trend of temperature. This test was performed using XLSTAT 2021 software. The positive value of Sen's slope shows an increasing trend and the negative value indicates a decreasing trend in the time series.

The flowchart of research methodology (Figure 2) depicts each step of the method used in the study.

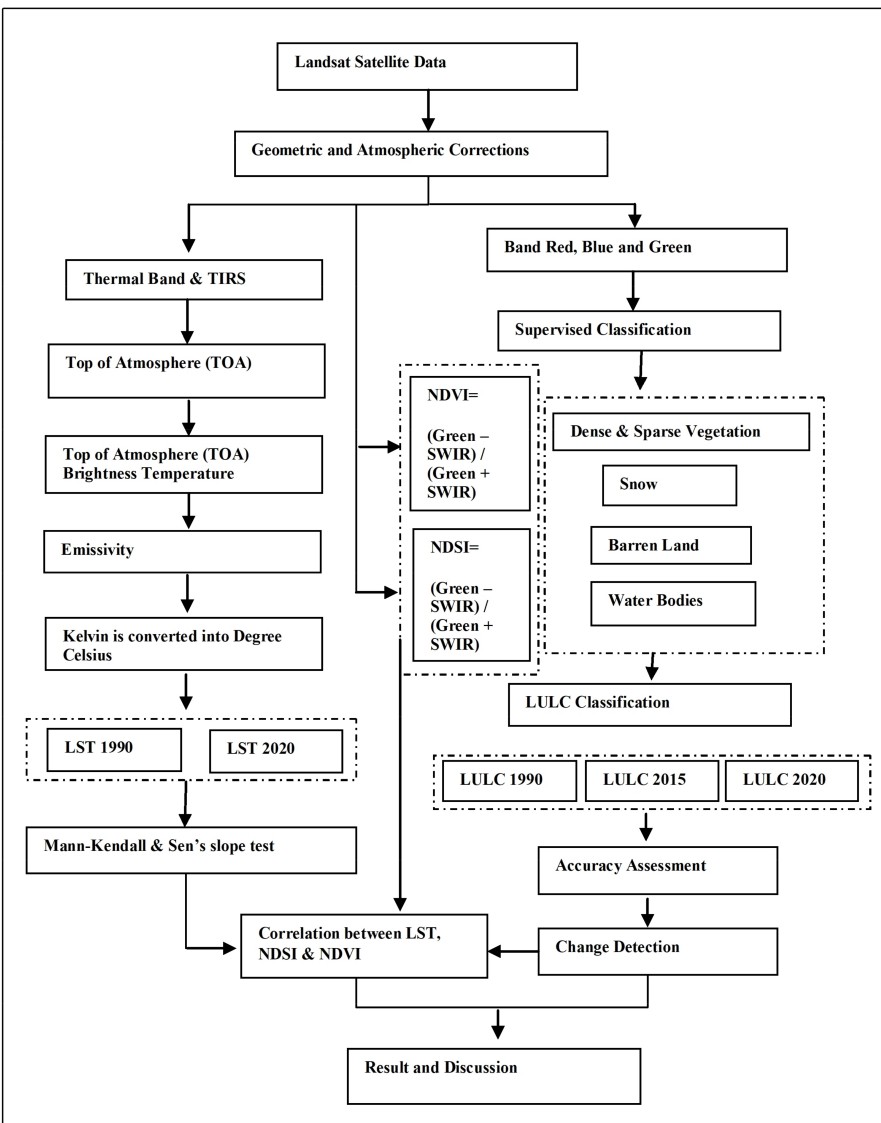

**Figure 2.** Flowchart of the Research Methodology.

*4.6. Accuracy Assessment*

The classification of satellite image is not completed until its accuracy is checked and the quality of the classification is evaluated by assessment of accuracy [36]. An error and confusion matrix technique is used to check the accuracy of the classified images. An error matrix is a very effective way to represent precision, as the precisions of each category are clearly described as well as the errors of inclusion (errors of commission) and errors of exclusion (errors of omission) present in the classification. Consequently, the overall accuracy, manufacturer and user accuracy, and Kappa statistic are calculated from the error matrix (Tables 3–5). The kappa coefficient is a measure of the proportion of improvement by the classifier over a purely random assignment to classes. Kappa accuracy is computed using the following equation.

$$K^n = (ND - P)/(N^2 - P)$$

where,

N = Total number of pixels;
D = Sum of correctly classified pixels;
P = Sum of product of row total and column total;
Note: Kappa of 0.85 means there is 85% better agreement than by chance alone.

**Table 3.** Error Matrix of Classified Image, 1990.

| Classified Data | Dense Vegetation | Sparse Vegetation | Snow | Barren Land | Water Body | Total | Commission Error (%) | User Accuracy (%) |
|---|---|---|---|---|---|---|---|---|
| Dense Vegetation | **120** | 5 | 0 | 2 | 0 | 127 | 5.5 | 94.48 |
| Sparse Vegetation | 3 | **130** | 0 | 13 | 0 | 143 | 7.80 | 90.90 |
| Snow | 0 | 0 | **105** | 8 | 2 | 115 | 8.69 | 91.30 |
| Barren Land | 1 | 4 | 3 | **125** | 4 | 136 | 8.08 | 91.91 |
| Water Body | 0 | 0 | 6 | 8 | **110** | 124 | 11.29 | 88.70 |
| Total | 124 | 134 | 114 | 154 | 116 | 518 | - | - |
| Omission Error (%) | 3.22 | 2.98 | 7.89 | 12.40 | 8.25 | - | - | - |
| Producer Accuracy (%) | 96.77 | 97.01 | 92.10 | 81.16 | 94.82 | - | - | - |

Overall Accuracy = 90.25%. Kappa Accuracy = 0.90.

**Table 4.** Error Matrix of Classified Image, 2015.

| Classified Data | Dense Vegetation | Sparse Vegetation | Snow | Barren Land | Water Body | Total | Commission Error (%) | User Accuracy (%) |
|---|---|---|---|---|---|---|---|---|
| Dense Vegetation | **150** | 6 | 0 | 4 | 0 | 160 | 6.25 | 93.75 |
| Sparse Vegetation | 4 | **125** | 0 | 10 | 0 | 139 | 10.07 | 89.92 |
| Snow | 0 | 0 | **110** | 6 | 4 | 120 | 8.33 | 91.66 |
| Barren Land | 2 | 5 | 4 | **120** | 3 | 134 | 10.44 | 89.55 |
| Water Body | 0 | 0 | 8 | 5 | **125** | 138 | 9.42 | 90.57 |
| Total | 156 | 136 | 122 | 145 | 132 | 691 | - | - |
| Omission Error (%) | 3.84 | 8.08 | 9.83 | 17.24 | 5.30 | - | - | - |
| Producer Accuracy (%) | 96.15 | 91.91 | 90.16 | 82.75 | 94.69 | - | - | - |

Overall Accuracy = 92.25%. Kappa Accuracy = 0.90.

The Kappa and overall accuracy were checked for the classified images of 1990, 2015, and 2020 which showed above the standard level accuracy (>0.85 and >85%). Kappa and overall accuracy of classified were obtained as 0.90 and 90.25% for 1990, 0.90 and 92.25% for 2015, and 0.91 and 91.45 for 2020, respectively. Producer accuracy or accuracy for each class and user accuracy were obtained for vegetation, snow, barren land, and water body, which showed a high level of accuracy (Tables 3–5). Furthermore, commission error and omission error were calculated. This indicates when an area is included in a category to

which it does not truly belong. Almost all accuracy measurement techniques were used to check the accuracy for all the classified images of the years 1990, 2015 and 2020.

**Table 5.** Error Matrix of Classified Image, 2020.

| Classified Data | Dense Vegetation | Sparse Vegetation | Snow | Barren Land | Water Body | Total | Commission Error (%) | User Accuracy (%) |
|---|---|---|---|---|---|---|---|---|
| Dense Vegetation | **160** | 8 | 0 | 3 | 0 | 171 | 6.43 | 93.56 |
| Sparse Vegetation | 6 | **182** | 0 | 9 | 0 | 191 | 4.71 | 95.69 |
| Snow | 0 | 0 | **125** | 5 | 3 | 133 | 6.01 | 96.74 |
| Barren Land | 4 | 4 | 3 | **170** | 5 | 182 | 6.59 | 93.40 |
| Water Body | 0 | 0 | 3 | 5 | **112** | 120 | 6.66 | 93.33 |
| Total | 170 | 186 | 131 | 189 | 120 | 626 | - | - |
| Omission Error (%) | 5.8 | 2.15 | 4.58 | 10.05 | 6.66 | - | - | - |
| Producer Accuracy (%) | 94.11 | 97.84 | 95.41 | 89.94 | 93.33 | - | - | - |

Overall Accuracy = 91.45. Kappa Accuracy = 0.91.

## 5. Result and Discussion

### 5.1. States of Land Use/Land Cover

Land Use and Land Cover (LULC) are often used simultaneously but they have two different meanings. Land cover includes the earth's features which are designed by nature, and people using those features are part of land use. Land use/land cover of the study area has been divided into five major categories for easy analysis. The result shows barren land and sparse vegetation are the major LULC types of the study area in which barren land occupies largest area among all the categories and sparse vegetation is second major LULC type. Barren land and sparse vegetation occupied an area of 844,244 ha. and 234,784 ha. in 1990, 834,544 ha. and 242,861 ha. in 2015, and 714,150 ha. and 279,228 ha. in 2020, respectively. Sparse vegetation cover is found in the form of grassland, scrub, mosses, and lichen in the study area. Grassland is generally used for animal grazing and it also provides an aesthetic view which attracts tourists from different parts of the world. Snow cover is third major LULC type and a one of the most important natural resources of the valleys; it fulfills the need for water of mountain dwellers. It covered an area of 188,445 ha., 183,080 ha., and 176,180 ha. in 1990, 2015, and 2020, respectively. The water received from snow or melting of glaciers is the only source of water because precipitation always occurs in the form of snowfall, and there is almost negligible rainfall. It is also very tough to fetch water from the rivers situated in the deep valleys. Thus, snow cover has a great significance on the lives of mountain dwellers of Lahaul and Spiti. Dense vegetation cover is the fourth major LULC type, which is generally found near streams and river channels. It only occupied an area of 33,864 ha., 35,682 ha., and 123,666 ha., in 1990, 2015, and 2020, respectively. It provides fodder for animal herds, fuel wood, and medicinal plants for mountain dwellers. Water body is ranked fifth and last position in all LULC types because this region is situated in a rain shadow area. The result extracted from Landsat image showing the status and spatial distribution of LULC has been given in (Table 6) and (Figure 3).

**Table 6.** States of Land Use Land Cover (ha.).

| LULC | 1990 | 2015 | 2020 |
|---|---|---|---|
| Total Area | 1,384,506 | 1,384,506 | 1,384,506 |
| Barren Land | 844,244 | 834,544 | 714,150 |
| Sparse Vegetation | 234,784 | 242,861 | 279,228 |
| Snow Cover | 188,445 | 183,080 | 176,180 |
| Dense Vegetation | 33,864 | 35,682 | 123,666 |
| Water Body | 83,169 | 88,339 | 91,282 |

Source: Calculated from Landsat Imagery.

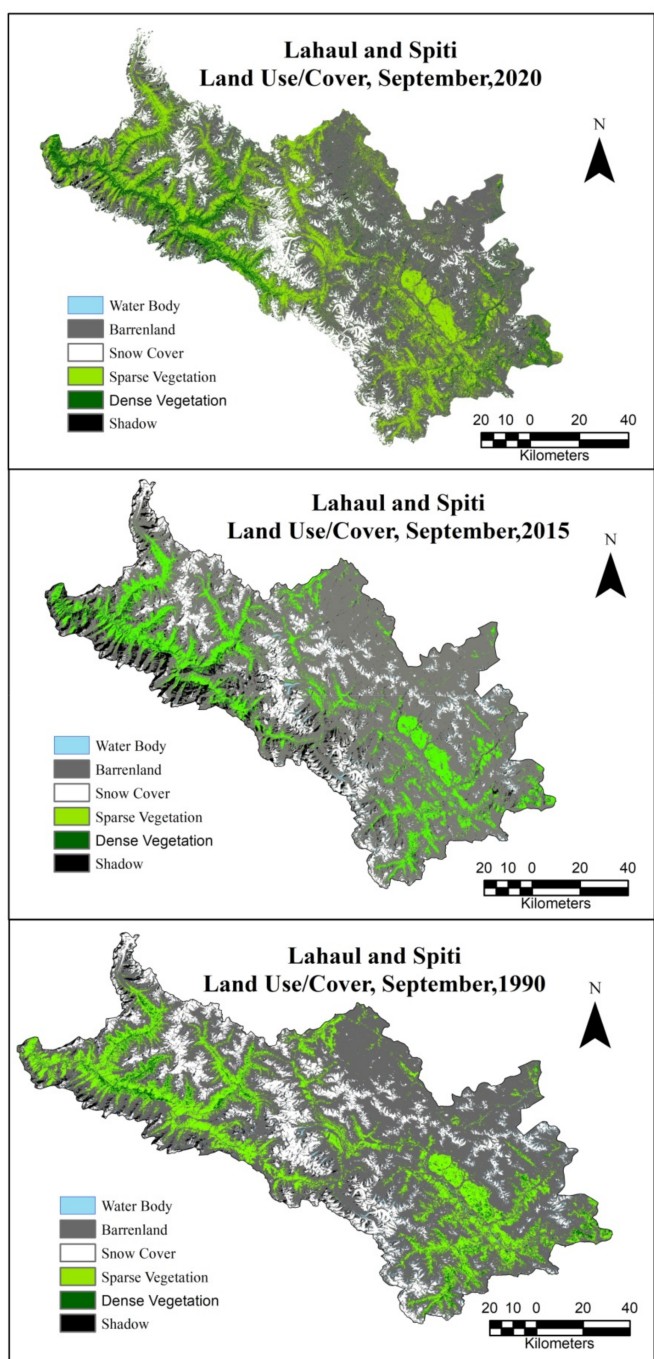

**Figure 3.** States of Land Use/Cover, 1990, 2015, and 2020.

As per the analysis, it was found that changes occurred in all the LULC types of Lahaul and Spiti during the research span. A major change took place in the area of dense vegetation cover. It saw an increase of 265 % registering an increase of 89,892 ha. from 1990 to 2020. This change occurred due to development of orchards farms in the form of small pockets.

### 5.2. Land Use/Cover Change Detection

Land Use/Land Cover change detection shows a significant change in all LULC types during the research span. Dense vegetation is not a common phenomenon in Lahaul and Spiti, but it increased 265% (+89,802 ha.) in the last three decades. Sparse vegetation cover also increased 18.92% (+44,444 ha.). The second significant change occurred in barren land,

which showed around a −15.40% (−130,094 ha.) decrease due to the increase in vegetation cover. The third major change has taken place in water bodies, with a 9.75% (+8113.4 ha.) increase in the area of water. The fourth major change occurred in snow cover, and this is the most active LULC type among all studied. The total snow cover decreased by −6.50% (−12,265 ha.) from 1990 to 2020 (Table 7). As per the analysis, an increase in vegetation cover and a decrease in snow cover is not a good sign for environmental health of the study area. A continued increase in vegetation cover can put the valley in the risk zone. Growing vegetation takes water from the ground and releases it into the atmosphere. Vegetation leaves also act as interceptors, trapping falling rain, which then evaporates and causes the rain to fall elsewhere, a process known as evapotranspiration. If evapotranspiration increases, precipitation will start taking place in the form of rain, and this not a good sign for Lahaul and Spiti because it is made of sedimentary rock. The increased rain can disturb the natural setup of the region and put it in a hazardous zone. On the other hand, increasing the density of vegetation can reduce the negative effects of rain on the land. The presence of dense vegetation can be considered as a balance factor in the stability of natural areas, especially since it has the ability to prevent erosion and possible mass movements. Declining snow cover area is another problem; it provides fresh water for all the purposes of mountain dwellers. Many snowcapped mountains have become naked mountains due to declining snowfall and upward shifts in the snowline and vegetation line positions. If it continues to decline the region would face the severe problem of water scarcity in the future.

**Table 7.** LULC Change Detection (1990–2020).

| LULC Types | 1990–2020 | |
| --- | --- | --- |
| | Area (ha.) | Area (in %) |
| Barren Land | −130,094 | −15 |
| Sparse Vegetation | +44,444 | 19 |
| Snow Cover | −12,265 | −7 |
| Dense Vegetation | +89,802 | 265 |
| Water body | +8113 | 10 |

Source: Computed from Landsat Imagery.

*5.3. Land Use/Cover Change Analysis through Matrix Table*

Matrix table analysis was used in the study to extract the exact value of LULC converted into other LULC types. It shows that 31% (435,217 ha.) of the area of Lahaul and Spiti experienced change and was converted into other LULC types from 1990 to 2020. The result matched with an earlier study. About a 2,151,647 ha (30%) area of Spiti valley has experienced change in the types of land cover in the 25 years from 1990 to 2015 [37]. In (Table 8), columns and rows represent the total sum of the amount of land for each LULC type between 1990 and 2020. The value in each cell of matrix table represents the amount of land that was converted from one LULC type to another. For example, the value of 31,944.4 ha. in the fourth column (Snow) and of the fourth row (Barren land) means that 31,944.4 ha area of snow cover was converted into barren land during the research span. The highlighted bold diagonal values indicate the area of each class that remains unchanged, while the off-diagonal values indicate the changed area for each LULC type. For instance, out of 33,861 ha. area of dense vegetation of 1990, 9937 ha. forest area remains unchanged in 2020.

**Table 8.** Land Use/Cover Matrix from 1990 to 2020.

| Reference Data | | | | | | | |
|---|---|---|---|---|---|---|---|
| | **1990** | | | | | | |
| | **LULC Types** | **Dense Vegetation** | **Sparse Vegetation** | **Snow** | **Barren Land** | **Water Body** | **Total** |
| **2020** | **Dense Vegetation** | **9937** | 32,834 | 10,841 | 57,316 | 12,688 | 123,614 |
| | **Sparse Vegetation** | 21,525 | **158,539** | 3986 | 90,377 | 4857 | 279,285 |
| | **Snow** | 0.40 | 66 | **132,948** | 35,032 | 6923 | 174,969 |
| | **Barren Land** | 1523 | 40,101 | 31,944 | **614,474** | 25,961 | 714,003 |
| | **Water Body** | 876 | 3207 | 8557 | 46,605 | **32,637** | 91,882 |
| | **Total** | 33,861 | 234,746 | 188,276 | 843,804 | 83,065 | 1,383,752 |
| | Major Changes | - | - | - | - | - | - |
| | No Change | - | - | - | - | - | - |

Source: Computed from Landsat Imagery.

*5.4. Land Use/Land Cover Change with Altitudinal Zones*

Lahaul and Spiti was classified into three different altitudinal zones (Figure 4) based on the Agro-Climatic Zones of the classification of Dhillon 1973, 1975 [38,39]. They include the lower altitudinal zone (2301–3000 m), alpine continental zone (3000–4250 m), and frigid continental zone (4250–6580 m). A significant change took place in the dense vegetation cover which increased in all the three altitudinal zones (Table 9). The maximum change occurred in the frigid continental zone, where +420% (+50,644 ha.) of the area of dense vegetation cover increased. Dense vegetation cover also increased in the alpine continental by +186% (+35,091 ha.) and the lower altitudinal zones +138% (+4021 ha.). Sparse vegetation decreased in the lower and alpine continental zones by −46% (−2641 ha.) and +186% (−11,154 ha.) but increased in the frigid continental zone by +50% (+58,336 ha.). A continued increase in vegetation cover area depicts that climate change is providing favorable conditions for the growth of vegetation. The increase in sparse vegetation only occurred in the frigid continental zone because snow/glacier retreated, and later on sparse vegetation grew on the bare ground in the forms of lichen and mosses. Snow cover was found in the alpine and frigid continental altitudinal zones but not in the lower altitudinal zone. Snow cover melts during the peak summer season and does not return until late post-summer because the lower altitudinal zone is situated at 2301 to 3000 m height.

The height of the lower altitudinal zone is not suitable for permanent snow cover in the study area. Furthermore, snow cover decreased in both alpine and frigid continental zones by −93% (−252 ha) and −7% (−13,650 ha), respectively. Another significant change occurred in water bodies, which decreased in all the three altitudinal zones (Figure 5). The maximum change took place in the alpine continental zone where −682% (−12,666 ha.) of water decreased; around −36% (−414 ha.) and −20% (−12,482 ha.) of the water decreased in lower and frigid zones, respectively, during the research span. The largest area of Lahaul and Spiti is occupied with barren land, and major changes in this category took place in the lower altitudinal zone. This clearly indicates that vegetation cover has increased at the cost of barren land.

**Table 9.** Land Use/Land Cover Change with Altitudinal Zones.

| | LULC Change in Lower Altitudinal Zone (2301–3000 m) | | | |
|---|---|---|---|---|
| **Zone -1 LULC** | **1990** | **2020** | **1990–2015** | |
| | **Area (ha.)** | **Area (ha.)** | **Area (ha.)** | **Area (%)** |
| Dense Vegetation | 2920 | 6941 | 4021 | 138 |
| Sparse Vegetation | 5684 | 3044 | −2641 | −46 |
| Snow Cover | - | - | - | - |
| Barren Land | 2726 | 928 | −1799 | −66 |
| Water Body | 1564 | 1150 | −414 | −36 |
| **Zone -2 LULC** | LULC Change in Alpine Continental Zone (3000–4250 m) | | | |
| Dense Vegetation | 18,890 | 53,981 | 35,091 | 186 |
| Sparse Vegetation | 113,158 | 102,004 | −11,154 | −10 |
| Snow Cover | 272 | 21 | −252 | −93 |
| Barren Land | 79,659 | 59,499 | −20,159 | −25 |
| Water Body | 14,522 | 1856 | −12,666 | −682 |
| **Zone -3 LULC** | LULC Change in Frigid Continental Zone (4250–6580 m) | | | |
| Dense Vegetation | 12,049 | 62,693 | 50,644 | 420 |
| Sparse Vegetation | 115,896 | 174,232 | 58,336 | 50 |
| Snow Cover | 188,172 | 174,522 | −13,650 | −7 |
| Barren Land | 761,147 | 653,332 | −107,815 | −14 |
| Water Body | 75,797 | 63,315 | −12,482 | −20 |

Source: Computed from Landsat Imagery.

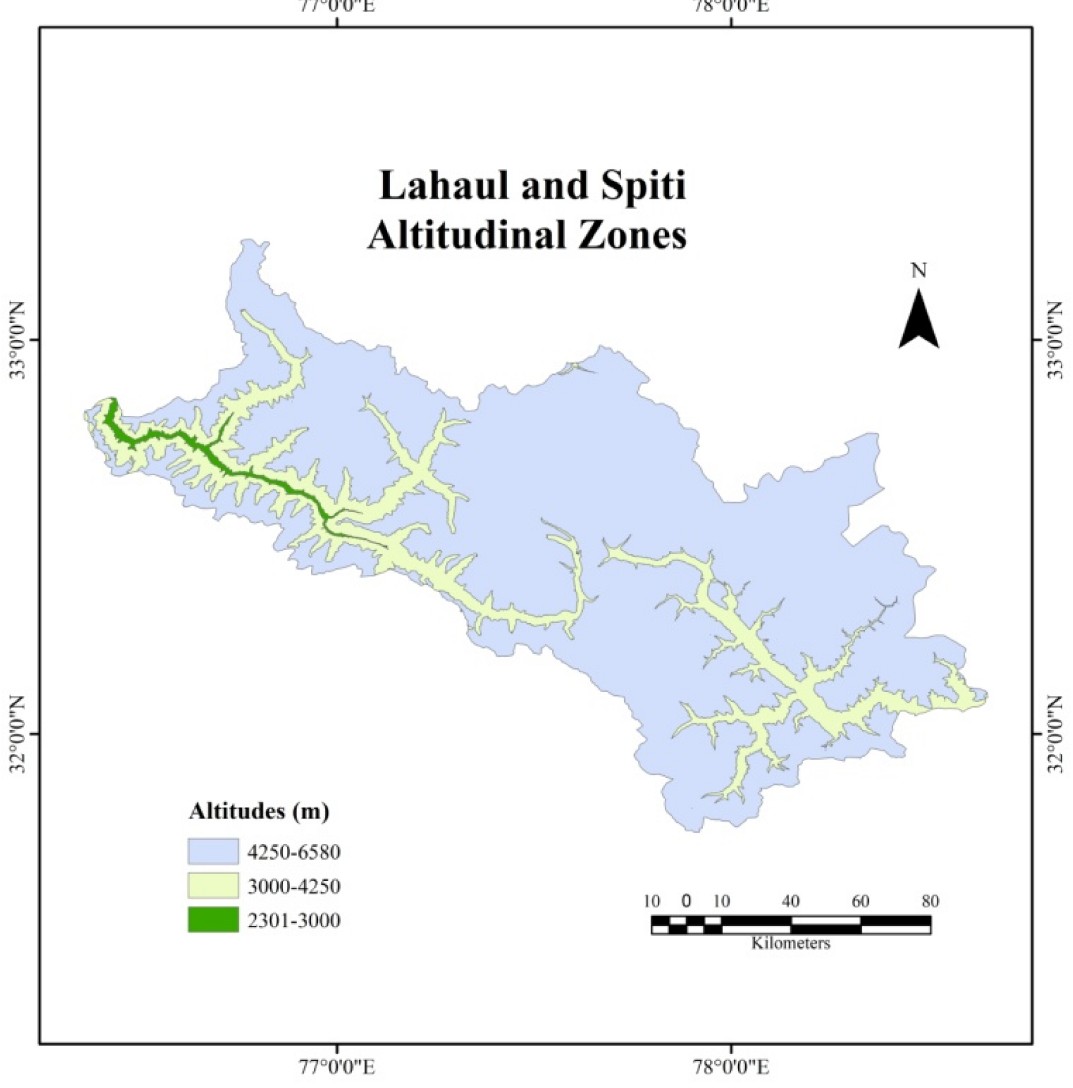

**Figure 4.** Altitudinal Zones of Lahaul and Spiti.

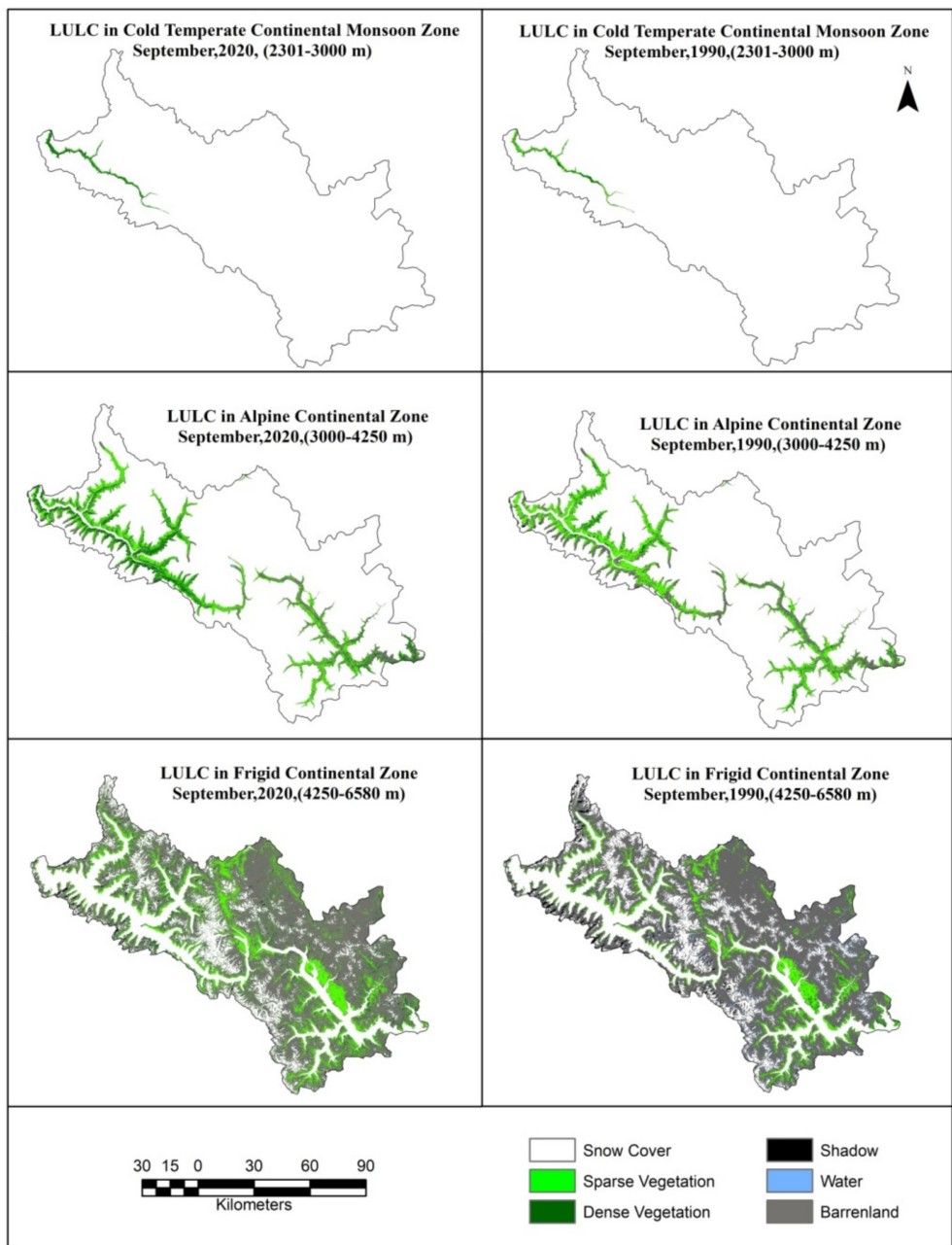

**Figure 5.** Land Use/Cover Change along Altitudinal Zones.

*5.5. Land Surface Temperature (LST) Change Analysis*

Land Surface Temperature (LST) determines the Earth's surface radiation and long-wave radiation energy balance, which are key input parameters in climate, hydrological, environmental, and biochemical models. Landsat data of September 1990 and 2020 was used to analyze the LST change. LST was retrieved using thermal band 6 of Thematic Mapper (TM) and bands 10 and 11 of Thermal Infrared Sensor (TIRS). In 1990, the maximum, minimum, and mean LST were 44 °C, −13 °C, and 15.5 °C, respectively (Table 10). Additionally, the maximum, minimum, and mean LST were 46 °C, −12 °C, and 17 °C, respectively, in 2020. The spatial distribution of the land surface temperature has been given in Figure 6 where LST is high in the lower elevation and it gradually decreases with increasing altitude of the valley.

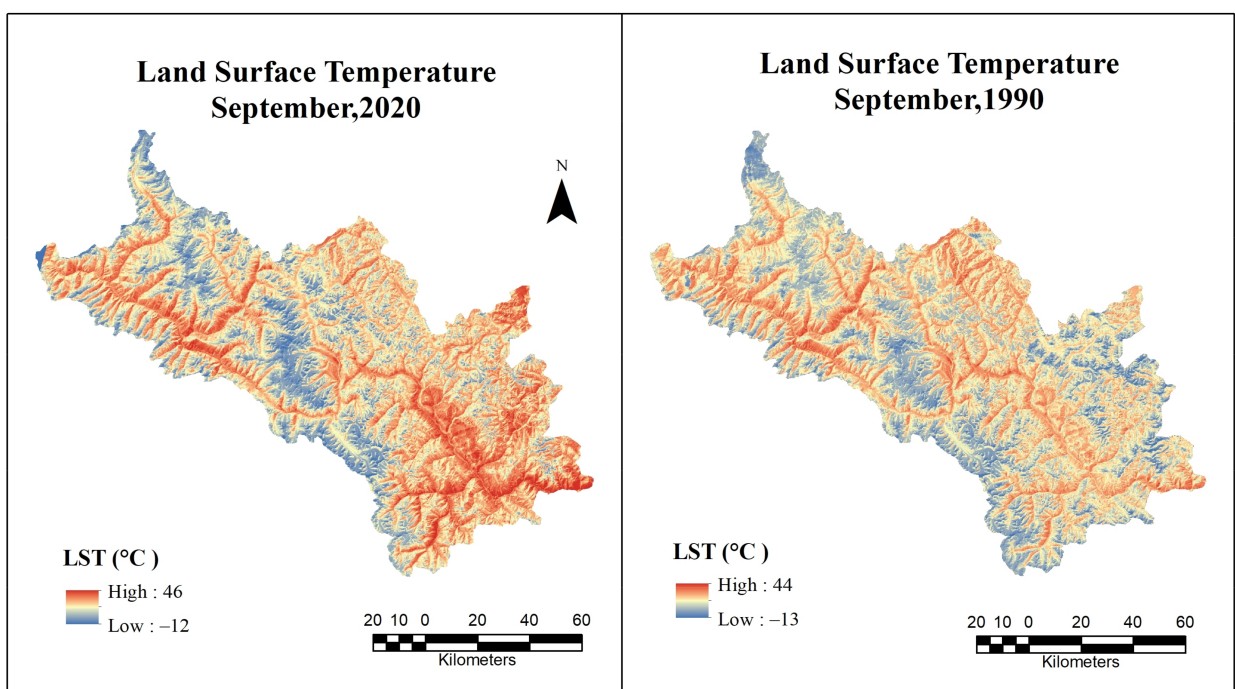

**Figure 6.** Spatial Distribution of Land Surface Temperature.

**Table 10.** LST Change from 1990 to 2020 (°C).

| Years | Minimum LST | Maximum LST | Mean LST |
|---|---|---|---|
| 1990 | −13 | 44 | 15.5 |
| 2020 | −12 | 46 | 17 |
| 1990–2020 | +1 | +2 | +1.5 |

Source: Computed from Landsat Imagery.

The minimum, maximum, and mean LST changed during the research span in the study area. The minimum, maximum, and mean LST increased by +1 °C, +2 °C, and +1.5 °C respectively. The increasing LST provides favorable conditions for vegetation growth, but it creates problems for areas with snow cover. This can be justified from the result of the LULC analysis, where vegetation cover increased and snow cover decreased in significant proportions. A continued increase in vegetation cover would enhance the heat trapping capability of the study area. The result of the LST change was validated with the help of the Mann–Kendall and Sen's slope trend tests. The trend of temperature in September was analyzed from 1981 to 2021. These two non-parametric tests are the best method to analyze trends in a metrological dataset. The descriptive statistics of temperature including maximum, minimum, mean, *SD*, and coefficient variation are given in (Table 11). The maximum, minimum, and mean temperatures were −1.47 °C, −5.49 °C, and −3.69 °C, respectively. The standard deviation (SD) and coefficient of variation of temperature were 0.859 °C and −23.24%, respectively.

**Table 11.** Summary of Statistical Techniques of Temperature (°C).

| Variable 1981–2021 | Minimum | Maximum | Mean | Std. Deviation | CV (%) |
|---|---|---|---|---|---|
| September | −5.49 | −1.47 | −3.697 | 0.859 | −23.24 |

Source: Computed by Author.

The result confirmed the increasing trend in temperature observed in the month of September (Figure 7) which is quite significant as obtained *p*-value was 0.011. Hence, the

null hypothesis H$_0$ was rejected and alternative hypothesis H$_1$ was accepted. The *p*-value of 0.11 indicated a significant increasing trend in temperature of the month of September in Lahaul and Spiti (Table 12).

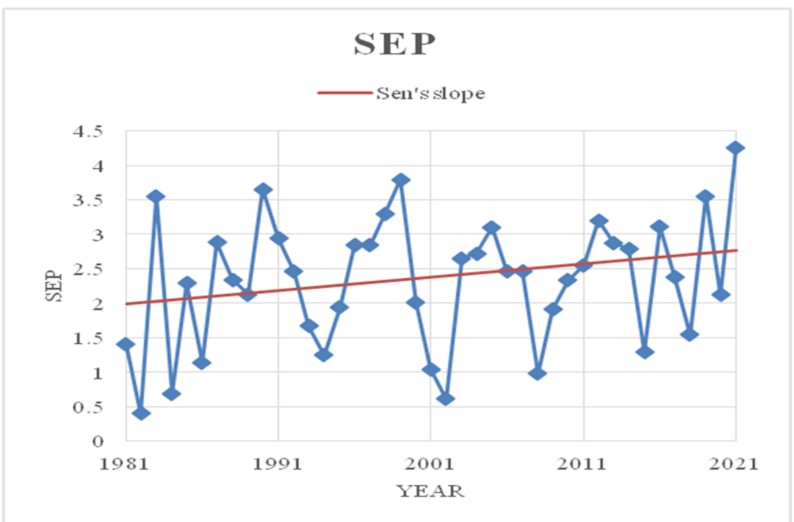

**Figure 7.** Monthly Trend of Temperature, 1981 to 2021.

**Table 12.** Mann–Kendall trend test/Two-tailed test.

| 1981–2021 | October |
|---|---|
| Kendall's tau | 0.276 |
| S | 226 |
| Var(S) | 7922.667 |
| *p*-value (Two-tailed) | **0.011** |
| Alpha | 0.05 |
| Sen's slope | 0.030 |

Source: Computed by Author.

*5.6. Impact of LULC Change on LST*

Land Use/Land Cover (LULC) change and Land Surface Temperature (LST) are interlinked and have significant impacts on each other [40,41]. The extent of this impact depends on the type and magnitude of the LULC change and the specific characteristics of the land surface and local climate [42–44]. Overall, the impact of the LULC change on LST is complex and depends on multiple factors. However, it is generally accepted that changes in land use and land cover can significantly alter local climate and temperature patterns, with important implications for human health, ecosystem services, and agricultural productivity. As per the LULC result, both dense and sparse vegetation have increased in the region. An increase in vegetation cover is a result of increasing temperature, which is providing suitable climatic conditions. However, vegetation typically has a cooling effect on the land surface by releasing moisture through transpiration, which helps to regulate temperature. The decline in snow cover area is a clear indication of rising of LST in Lahaul and Spiti. There was an increase of around 10% in water body from 1990 to 2020. This change has occurred because of the development of new glacial lakes. Furthermore, the study investigated how the surface temperature of each LULC type changed over time. The mean surface temperature of each LULC type was extracted by averaging all the sample pixels of the classified LULC types. The mean STC of each LULC type i.e.dense vegetation, sparse vegetation, snow cover, water body, and barren land was 16 °C, 17 °C, −3 °C, 19 °C and 5 °C in 1990, and 19 °C, 18 °C, −2 °C, 22 °C and 8 °C respectively in 2020. The change in surface temperature of each LULC types can be easily seen in Figure 8. The mean

surface temperature of dense vegetation, sparse vegetation, snow cover, water body, and barren land increased by +3 °C, +1 °C, +1 °C, +3 °C, and +3 °C, respectively, in the last three decades. The most active and susceptible to temperature is snow cover among all land cover types. A 1 °C increase in surface temperature of snow cover areas could lose substantial amount of snow from the area mainly during summer time.

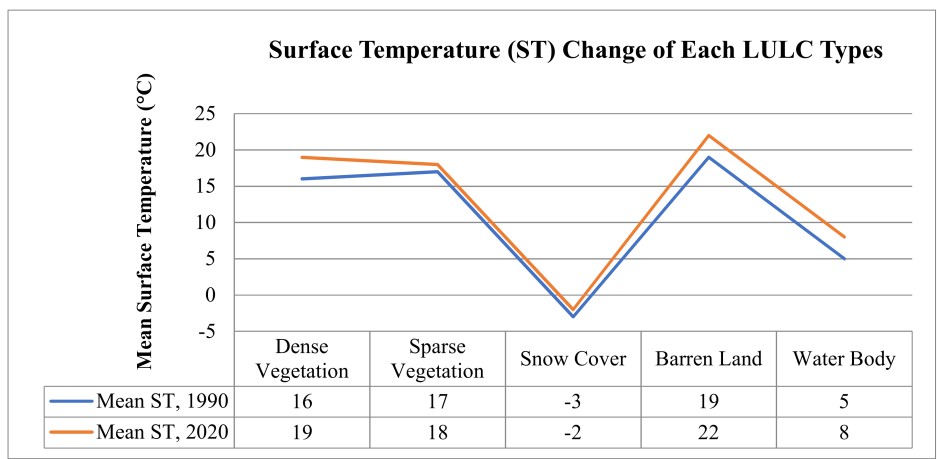

**Figure 8.** Surface Temperature (ST) Change of Each LULC Types.

Overall, changes in land use and land cover can have complex effects on LST, and the specific impacts will depend on a range of factors, including the type and extent of land use change, the climate of the area, and the season. Understanding these relationships is important for predicting and mitigating the impacts of land use change on temperature and climate.

### 5.7. The Relationship of LST with NDSI and NDVI

The relationship was established on the basis of the pixels values extracted using the fishnet technique from Land Surface Temperature (LST), Normalized Difference Vegetation Index (NDSI), and Normalized Difference Vegetation Index (NDVI) The fishnet tool creates a feature class that contains a net of rectangular cells. There are three basic sets of information to create fishnet: the spatial extent of fishnet, the number of rows and columns, and the angle of rotation (Figure 9). Karl Pearson's coefficient of correlation was used to examine the relationship of LST with NDSI and NDVI. LST was considered an independent variable and NDSI as well as NDVI were taken as dependent variables in this calculation.

The result of correlation of coefficient of LST with NDVI shows a positive correlation of $R^2 = 0.47$ in 1990 and $R^2 = 0.61$ in 2020 (Figure 10). The climatic conditions of Lahaul and Spiti are not suitable for vegetation, but the change in regional temperature due to global warming creates ideal conditions for vegetation growth. The result of correlation of coefficient clearly indicates that LST is providing favorable environmental conditions for vegetation growth.

On the other hand, the analysis of the correlation coefficient of LST with NDSI shows a negative correlation $R^2 = 0.3827$ in 1990 and $R^2 = 0.4285$ in 2020. An increase in the value of $R^2$ (correlation coefficient) indicates that a negative correlation is becoming strong between LST and NDSI. It confirms that rising LST has a negative impact on snow cover area in Lahaul and Spiti. This will result in a significant decrease in snow cover in alpine and frigid continental zones, where precipitation will tend to fall in the form of rain and there could also be more rain-on-snow events. This could result in flash floods.

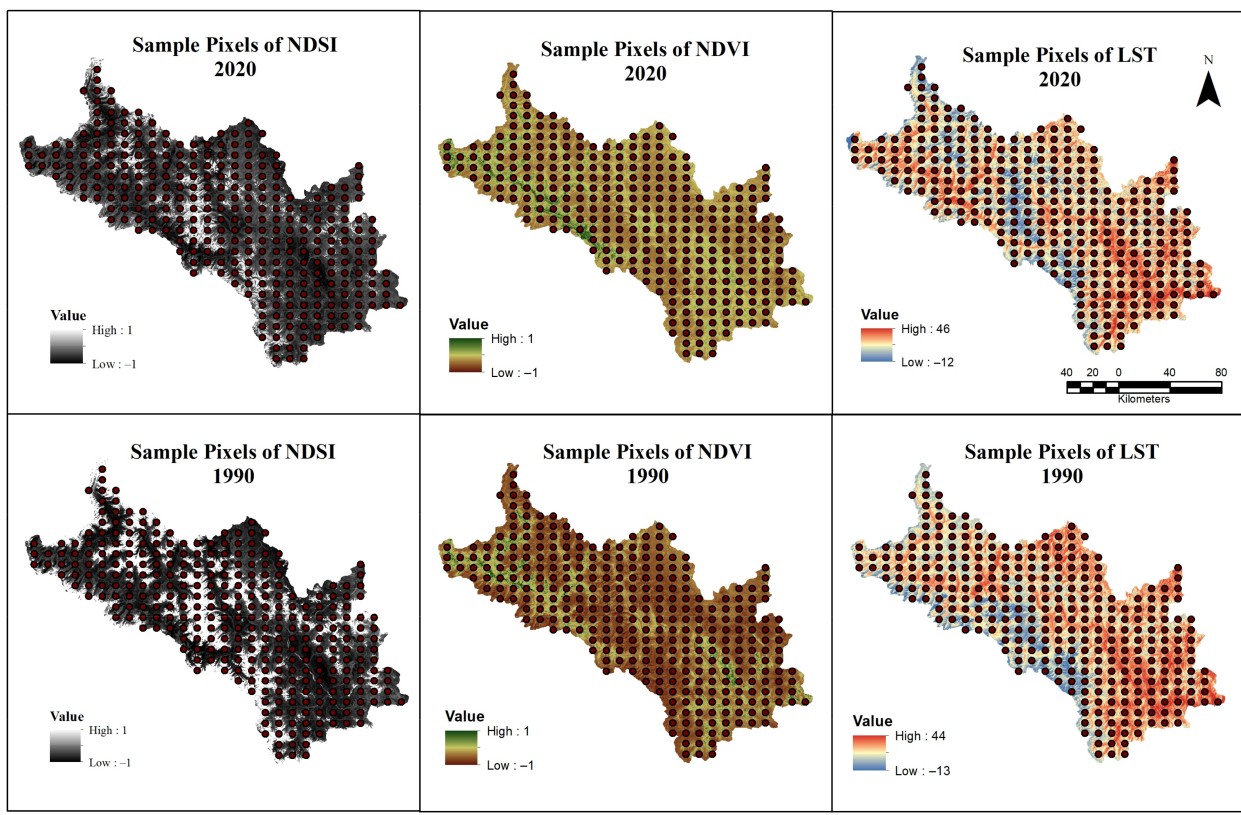

**Figure 9.** The Sample Pixels of LST, NDVI, and NDSI.

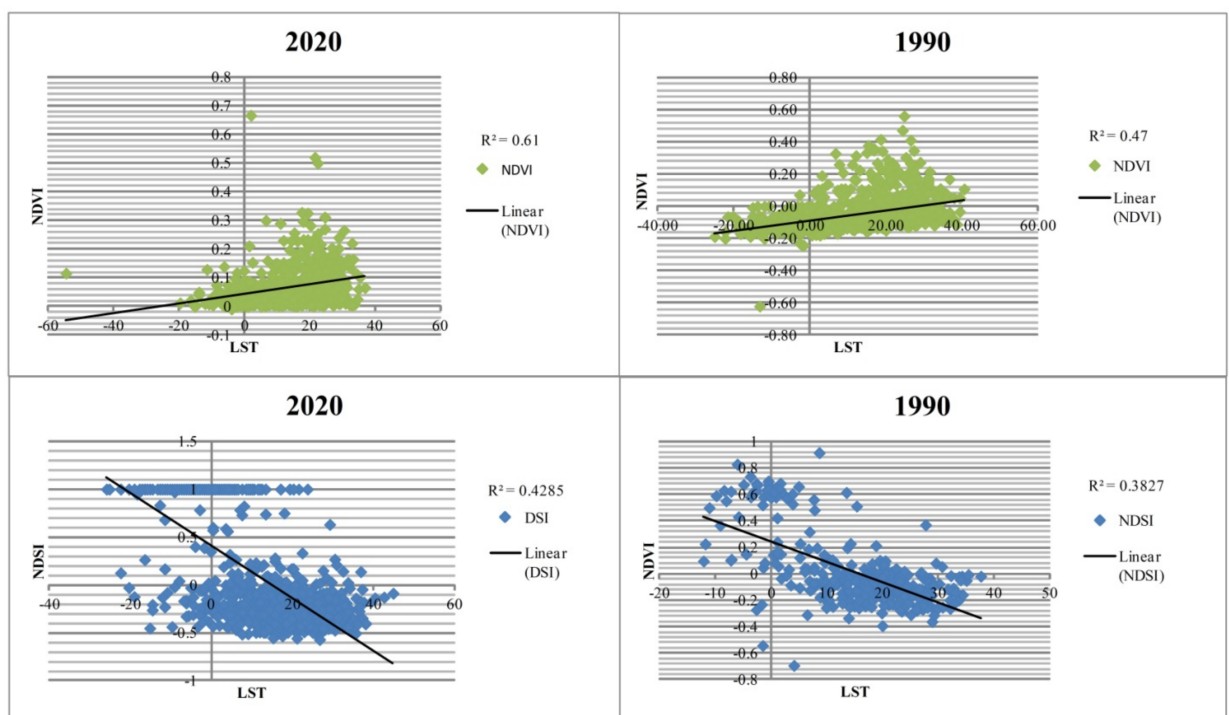

**Figure 10.** The correlation of coefficient of LST with NDVI and NDSI.

## 6. Conclusions

Based on the Landsat images of 1990, 2015, and 2020, LULC changes and its impact on LST in Lahaul and Spiti district was examined. The study area has been classified into five LULC types, for example barren land, sparse vegetation, snow cover, water body, and dense

vegetation. A significant change occurred in vegetation cover, where dense and sparse vegetation increased. Water body area increased by 10%, which is in contrast with the global trend. Conversely, snow cover decreased by −7% during the research span. The region is a rain shadow area where glacier/snowmelt water is the single source of fresh water. Thus, the increase in water body is the result of the conversion of glacier/snow cover area into several glacial lakes. The Lahaul and Spiti region will face a severe problem of water scarcity if snow cover continues to decrease at this rate. The livelihood of the mountain dwellers depends upon the two most important LULC types, vegetation and snow cover. Analysis of the LST indicated that the minimum, maximum, and mean LST increased by +1 °C, +2 °C, and +1.5 °C, respectively. The result of the LST change was validated with the help of the Mann–Kendall and Sen's slope trend test. It also shows a significant increasing trend in temperature in the month of September (*p*-value 0.011). LULC change has significant impacts on Land Surface Temperature (LST). The effects of LULC change on LST can be seen in the form of vegetation increase, deglaciation, changes in agricultural practices, and alteration of wetlands. These changes in LST can have far-reaching effects on the local and regional climate, as well as on ecosystems and human communities. It is important to understand these relationships in order to effectively manage and mitigate the impacts of LULC change on LST and the associated environmental, social, and economic impacts. Further research is needed to fully understand the complex interactions between LULC change and LST and to develop effective strategies for managing land use in a way that balances human needs with environmental sustainability.

**Author Contributions:** Conceptualization, M.A.H. and P.K.; methodology, M.A.H. and P.K.; software, M.A.H.; validation, M.A.H.; formal analysis, M.A.H., P.K. and B.G.; investigation, M.A.H. and P.K.; resources, M.A.H.; data curation, M.A.H. and P.K.; writing–original draft, M.A.H.; writing–review & editing, P.K. and B.G.; supervision, P.K. All authors have read and agreed to the published version of the manuscript.

**Funding:** Authors acknowledge DST BRICS 087 Research Project Dy. No. C/1957/IFD2020-21 sanctioned by International Cooperation Division of Department of Science and Technology (DST), Government of India (GoI) and ICSSR Major Research Project grant sanction No. 02/13847/OBC/2021-22/ICSSR/RP/MJ and Faculty Research Programme Grant–IoE Ref. No./IoE/2021/12/FRP for financial support and University of Delhi to facilitate logistics support to conduct search work.

**Data Availability Statement:** Data were acquired from Earth Explorer, USGS and the NASA Langley Research Center POWER Project. Primary data was collected through field survey. https://earthexplorer.usgs.gov/ (accessed on 22 February 2023), https://power.larc.nasa.gov/ (accessed on 22 February 2023).

**Conflicts of Interest:** The authors declare no conflict of interest.

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
