# Peer review of "Assessment of Spatio-Temporal Land Use/Cover Change and Its Effect on Land Surface Temperature in Lahaul and Spiti, India"

_land, doi:10.3390/land12071294_

Round 1
Reviewer 1 Report (New Reviewer)
The effort of analysis of spatio-temporal LULCC in India and determining its effect on LST is interesting and will bring a significant contribution in this field.
Besides this, the manuscript needs serious improvements.
Introduction section seems small and should be increased. Current trends in LULCC analysis should be presented.
L102. What preprocessing steps were made?
Table 1. Clarify the purpose of using false color composite (4,3,2) instead of true-color (3,2,1)? Same for L126. TCC is (3,2,1) for Landsat 5 and 7 while FCC – (4,3,1) (https://custom-scripts.sentinel-hub.com/custom-scripts/Landsat-57/composites/)
L129. Spectral signatures plot in this case should be presented
Equations should be presented according to MDPI rules.
References in subsections 4.2-4.5 should be added.
L241. There is no link for Fig. 2 in main text
References part is too small. Authors should make deeper literature analysis
I wish that my comment would be helpful in improving the quality of this research.
Thank you.
Author Response
To,
The Editorial
Land
We would like to thanks reviewer for making efforts and giving valuable comments and suggestions. All the suggestions and comments have been incorporated in the manuscript. Response of each comment has been given below:
Comment1. Introduction section seems small and should be increased. Current trends in LULCC analysis should be presented.
Comment 2. L102. What preprocessing steps were made?
Comment 3. Table 1. Clarify the purpose of using false color composite (4,3,2) instead of true-color (3,2,1)? Same for L126. TCC is (3,2,1) for Landsat 5 and 7 while FCC – (4,3,1) (https://custom-scripts.sentinel-hub.com/custom-scripts/Landsat-57/composites/)
Response: False color composites have been used in the present study because it allows us to visualize wavelengths that the human eye cannot see (i.e. near-infrared). Using bands such as near infra-red increases the spectral separation and often increases the interpretability of the data.
Comment 4. L129. Spectral signatures plot in this case should be presented
Response: Spectral signature has been taken as sample for each class of LULC and it is a process to classify the image. The final map of LULC has been given in the manuscript.
Comment 5. Equations should be presented according to MDPI rules.
Response: We regret to say that we were not able to find MDPI rules for equation. It would be highly appreciated if editor provide us some samples of it.
Comment 6. References in subsections 4.2-4.5 should be added.
Response: References in subsections 4.2 for NDSI and NDVI have been added as
Valovcin, F. R., (1978) “Spectral radiance of snow and clouds in the near infrared spectral region”, AFGL-TR-78-0289, ADA 063761.
Valovcirh F. R., (1976) “Snowicloud discrimination”, AFGL-TR-76- 0174, ADA 032385.
Jiang, Z., et al. (2006) Analysis of NDVI and scaled difference vegetation index retrievals of vegetation fraction. Remote sensing of Environment,. 101(3): p. 366-378.
Comment 7. L241. There is no link for Fig. 2 in main text
Response: Figure 2 has been mentioned in the text.
Cpmment 8. References part is too small. Authors should make deeper literature analysis
Response: Agreed with referee, authors have added more references in the manuscript.

Reviewer 2 Report (New Reviewer)
· Abstract part requires complete change with a clear hypothesis and results.
· Introduction part should be modified indicating the hypothesis of the study, study gap and how this study will address the issue.
· Conclusion is very lengthy- can be made clear and short.
· From the correlation results, it seems non-significant relationship which can be mentioned rather than quantitative term like moderate/ strong etc.
· The primary aim of the study i.e. establishing the relationship between change in land use and land surface temperature- which is poorly discussed. Much emphasis should be given in this particular area since it is an interesting part of the study.
· The result and discussion can be improved by incorporating suitable references.
Author Response
To,
The Editorial
Land
We would like to thanks reviewer for making efforts and giving valuable comments and suggestions. All the suggestions and comments have been incorporated in the manuscript. Response of each comment has been given below:
Referee 2
Comment 1. Abstract part requires complete change with a clear hypothesis and results.
Response: The present study is not hypothetical based study. It is a objective based study where the main objective of the study was to analyze Spatio-temporal change in Land Use/Cover (LULC) & its effect on Land Surface Temperature (LST) and to establish a correlation of LST with Normalized Difference Vegetation Index (NDVI) & Normalize Difference Snow Index (NDSI). Other objective was to analyze the trend of temperature in Lahual and Spiti.
Comment 2. Introduction part should be modified indicating the hypothesis of the study, study gap and how this study will address the issue.
Response: As this is an objective based study, authors have not taken any hypothesis. It has been discussed in the background of the study area that Lahaul and Spiti district is a cold desert mountain which is situated in the trans-Himalaya region. This region is not easily accessible for human beings and very few studies have been made on LULC and LST. It becomes possible with the development of geospatial technique. So, there is a huge gap in the study and a continued geospatial monitoring the region fill the gap in the study. This study can help in a making government program and policy to resolve the issues of mountain dwellers of Lahaul and Spiti.
Comment 3. Conclusion is very lengthy- can be made clear and short.
Response: Agreed with referee and conclusion has been shorten.
Comment 4. From the correlation results, it seems non-significant relationship which can be mentioned rather than quantitative term like moderate/ strong etc.
Response: Agreed with referee, moderate and strong have been removed from discussion.
Comment 5.The primary aim of the study i.e. establishing the relationship between change in land use and land surface temperature- which is poorly discussed. Much emphasis should be given in this particular area since it is an interesting part of the study.
Response: Agreed with referee, impact of LULC change on LST has been discussed in details.
Comment 6. The result and discussion can be improved by incorporating suitable references.
Response: Agreed with referee and the result and discussion have been improved.

Round 2
Reviewer 2 Report (New Reviewer)
Dear authors, the manuscript has been improved from its original version. I could find minor mistake please check it out.
Good wishes
Line 17: Check spelling of analyse
Line 35: Oinam et al., 2005
Figure 2: Check out the figure. Part of the figure is missing- may be alignment issue
Line 527-528: check the sentence
References are not in uniform format. Authors may check the journal format and update the references
Author Response
2nd Review
To,
The Editorial
Land
We would like to thanks reviewer for making efforts and giving valuable comments and suggestions. All the suggestions and comments have been incorporated in the manuscript. Response of each comment has been given below:
Comment1: Line 17: Check spelling of analyse
Response: Agreed with referee and correction has been made.
Comment 2: Line 35: Oinam et al., 2005
Response: Agreed with referee and correction has been made.
Comment 3: Figure 2: Check out the figure. Part of the figure is missing- may be alignment issue
Response: Agreed with referee and the alignment of figure has been corrected.
Comment 4; Line 527-528: check the sentence
Response: Agreed with referee and the sentence from line 527-528 has been reframed.
Comment 5: References are not in uniform format. Authors may check the journal format and update the references
Response: Agreed with referee and references have been listed as per journal format.

This manuscript is a resubmission of an earlier submission. The following is a list of the peer review reports and author responses from that submission.
Round 1
Reviewer 1 Report
This manuscript studied the spatio-temporal LULC change and its effect on LST using NDVI and NDSI derived from Landsat images. It is an interesting work in the field of land. However, some other problems in the manuscript are still concerned in the following:
1. The Landsat images in 1990, 2000, 2020 are used in this study. In order to make a comprehensive evaluation, the images in 2010 should be included.
2. The introduction is too simple. Please show more information.
3. The mathematical equations should be numbered.
4. In Lines 132 133, the authors said that NDVI is computed by visible and near infrared bands. In fact, NDVI is derived from red and near infrared bands, please refer to the review article “High quality vegetation index product generation: A review of NDVI time series reconstruction techniques”.
5. Only 13 references are cited. More related works are suggested.
6. The conclusion could be shortened.

Reviewer 2 Report
The study was to analyze Spatio-temporal change in Land Use/Cover (LULC) & its effect on Land Surface Temperature (LST) in Lahaul and Spiti, India. This is an interesting topic. However, this study was not innovative enough.
So I suggest this manuscript should be rejected.
Other issues are as follows.
(1)The introduction does not provide a detailed description of the significance of the study and the progress of previous studies.
(2)The manuscript used Sobrino’s formula, LSE= 1.0094+0.047*Ln (NDVI), to get the LSE of the study area. I think the empirical formula should vary from region to region. The authors should prove that this formula is appropriate in the study area or change correlation coefficients.
(3)All references cited in the text should be given in the reference list and vice versa. For example, the reference y (Sobrino and Raissouni, 2001) can not be found in the bibliography.The format of the reference document is in a mess.
(4)The manuscript stated that the study area is very sensitive to anthropogenic activities and climate change and it should not be changed at any cost. The tone is too absolute, generally be ruled out. Change is the only absolute in the world. We can not stop it.
(5)The picture in the top right corner of Figure 1 can be removed.
Reviewer 3 Report
The article has serious flaws, which make it inadequate for publication. Although it maps the change of LULC clearly, using a snapshot of LST at certain hours and dates is not trustworthy. Furthermore, the statistical model, i.e. use of correlation or different LULC at different times, is weak, given that it does not account for (1) collinearity between the independent variables, (2) does not account for temporal change, and (3) does not include proper control variables. A proper model was needed to account for the fixed and random effects. This lead to the most troublesome component of the article: scientific fallacies, i.e. generalization without evidence and interpreting correlation as causation. The conclusion of the paper, for instance, suggests that “it can be concluded that the increase in global temperatures and the temperature increases in the Himalayas is the most important factors in the reduction of snow-covered areas in the study area” and “This increase causes an increase in vegetation and water bodies“. This study does not offer evidence for such conclusions. The article in this current shape is not adequate for publication. The reviewer suggests that the authors use the materials and mappings of LULC changes to write a brand-new manuscript in future.